## Research Article

# B-cell targeting with anti-CD38 daratumumab: implications for differentiation and memory responses

Dorit Verhoeven[1,2] , Lucas Grinwis[2], Casper Marsman[3], Machiel H Jansen[2], T2B Consortium*, Ester MM Van Leeuwen[2], Taco W Kuijpers[1,2]

B cell–targeted therapies, such as CD20-targeting mAbs, deplete B cells but do not target the autoantibody-producing plasma cells (PCs). PC-targeting therapies such as daratumumab (anti-CD38) form an attractive approach to treat PC-mediated diseases. CD38 possesses enzymatic and receptor capabilities, which may impact a range of cellular processes including proliferation and differentiation. However, very little is known whether and how CD38 targeting affects B-cell differentiation, in particular for humans beyond cancer settings. Using in-depth in vitro B-cell differentiation assays and signaling pathway analysis, we show that CD38 targeting with daratumumab demonstrated a significant decrease in proliferation, differentiation, and IgG production upon T cell–dependent B-cell stimulation. We found no effect on T-cell activation or proliferation. Furthermore, we demonstrate that daratumumab attenuated the activation of NF-κB in B cells and the transcription of NF-κB–targeted genes. When culturing sorted B-cell subsets with daratumumab, the switched memory B-cell subset was primarily affected. Overall, these in vitro data elucidate novel non-depleting mechanisms by which daratumumab can disturb humoral immune responses. Affecting memory B cells, daratumumab may be used as a therapeutic approach in B cell–mediated diseases other than the currently targeted malignancies.

## Introduction

An essential process of humoral immunity is B-cell differentiation into antibody-producing plasma cells (PCs) (1). B cells can be activated through T cell–dependent (TD) activation, provided as help from T-follicular helper cells via CD40–CD40 ligand (CD40L) engagement, or through T cell–independent (TI) manners via TLR9 stimulation (1, 2). After activation, B cells are able to proliferate and differentiate into plasmablasts (PBs). Dependent

on the activating conditions, B cells differentiate further into immunoglobulin-producing PCs or become memory B cells, which can respond rapidly upon subsequent encounter of cognate antigen (3). The cell surface molecules IgD, CD19, CD20, CD27, CD38, and CD138 are frequently used to identify the main B-cell populations in peripheral blood (4). The role of paired box 5 (PAX5), NF-κB, B lymphocyte–induced maturation protein-1 (BLIMP1), and interferon regulatory factor 4 (IRF4) as major drivers of B-cell identity and PC differentiation has been well established (1, 4, 5). In contrast, the mechanisms restricting PC differentiation remain incompletely understood.

Derailed B-cell function and PC generation is believed to play a key role in the pathogenesis of autoimmune disorders, such as systemic lupus erythematosus (6). A small fraction of autoimmune patients remains unresponsive to conventional B cell–depleting mAbs directed against CD20, where it is hypothesized that autoreactive PBs (CD20⁻CD38⁺) or PCs (CD20⁻CD38⁺CD138⁺) differentiate into long-lived PCs and reside in the bone marrow or inflamed tissues, where they are not depleted by these therapies. CD38-expressing malignant B cells and long-lived PCs can be targeted by novel B cell–targeted therapies such as the anti-CD38 mAbs daratumumab (DARA, trade name Darzalex) or isatuximab (trade name Sarclisa), which are currently approved for treatment of multiple myeloma (MM) (7, 8, 9, 10). These antibodies are highly efficacious and safe in MM patients. In MM patients, anti-CD38 therapy is associated with decreased immunoglobulin levels in serum, reduced autoantibody levels, increased frequency of infections, and reduced vaccination responses (to SARS-CoV-2) (8, 9, 11, 12, 13, 14, 15). However, it should be noted that these patients have altered function of the immune system induced by the disease itself and are heavily pretreated with other immunomodulatory drugs too (16). The mechanisms underpinning how anti-CD38 therapy influences normal PCs or PC differentiation beyond cancer settings have remained virtually unexplored.

CD38 has extensively been used to classify various lymphocyte subsets in humans and mice, as an activation marker or biomarker associated with poor prognosis in MM (17). CD38 is a multifunctional

---

[1]Amsterdam UMC, University of Amsterdam, Department of Pediatric Immunology, Rheumatology and Infectious Diseases, Emma Children's Hospital, Amsterdam, The Netherlands  [2]Amsterdam UMC, University of Amsterdam, Department of Experimental Immunology, Amsterdam Institute for Infection and Immunity, Amsterdam, The Netherlands  [3]Sanquin Research and Landsteiner Laboratory, University of Amsterdam, Department of Immunopathology, Amsterdam, The Netherlands

Correspondence: d.verhoeven1@amsterdamumc.nl
*T2B Consortium members are listed in the below Appendix.

transmembrane glycoprotein possessing both enzymatic and receptor functions. Topologically, CD38 can behave as a type II or type III membrane protein depending on the orientation of the catalytic domain ([18], [19], [20]). Most commonly, the catalytic domain is situated in the extracellular compartment (type II). Given CD38's multiple possible orientations and enzymatic functions, its substrate and products would be consumed or produced in the extracellular or intracellular compartment. The enzymatic functions of CD38 include the conversion of NAD$^+$ into ADP-ribose (ADPR) and nicotinamide (NAM). Secondarily, it degrades NAD$^+$ via cyclase activity resulting in cyclic ADPR (cADPR), which results in increased Ca$^{2+}$ mobilization, shown by enzymatic assays of human CD38 ([20], [21]). Also, CD38 can metabolize NAD precursors and therefore regulates extracellular NAD$^+$ availability, as shown in CD38 knockout mice ([22], [23]). Hereby, CD38 may influence activation of NAD$^+$-dependent enzymes known to be involved in the canonical NF-κB pathway activation ([24], [25]). Besides this, CD38 is able to interact with CD31 to induce adhesion to endothelial cells ([26]). In B cells, activating CD38 mAbs have been shown to lower the threshold for B-cell receptor (BCR)–mediated B-cell activation ([27]). Furthermore, it has been shown in vitro that targeting CD38 with daratumumab, or removing CD38 with CRISPR/Cas9, inhibits the association of CD19 with the BCR, impairing BCR signaling in normal and malignant human B-cell lines ([28]).

Because daratumumab is known to interfere with B-cell activation in cell lines in vitro and because MM patients on daratumumab treatment show reduced immunoglobulin production in vivo, the effect of daratumumab may be beyond B-cell depletion only. Therefore, we focused in this study on the effects of daratumumab on the functional characteristics of B cells using in-depth in vitro B-cell differentiation assays and signaling pathway analysis.

# Results

## B-cell stimulation in the presence of daratumumab results in decreased B-cell proliferation upon TD stimulation

To mimic the in vivo situation in lymphoid organs where B and T cells are exposed to daratumumab (hereafter referred to as DARA), we used a previously established B-cell differentiation assay and added different concentrations of DARA ([29], [30]). In short, CFSE-labeled PBMCs were corrected for 25,000 B cells per well and subsequently cultured for 6 d in the presence of TI (CpG ± IL-2) or TD (αCD40 + IL-21 ± αIgM) stimuli (Fig 1A). Different concentrations DARA, for example, 0, 0.1, 1.0, 10, or 100 µg/ml, were added at the start of culture based on pharmacological in vivo studies ([31]). The addition of DARA did not change the percentages of B cells in the unstimulated and stimulated conditions (Figs 1B and S1A). A small decrease in total lymphocyte percentages was observed only in the conditions with 10 and 100 µg/ml DARA in the CpG + IL-2–stimulated conditions (Fig S1B). After stimulation with αCD40 + IL-21 (with and without αIgM), B cells displayed a dose-dependent decrease in proliferation in the presence of DARA (Fig 1C). Characteristically, the activation of B cells with these stimuli for 6 d results in

consecutive cycles of cell division accompanied by the loss of IgD and the up-regulation of PB markers CD27 and CD38. The decrease in proliferation in the TD-stimulated conditions resulted in a significant increase in percentages of double-negative (IgD$^-$CD27$^-$) B cells with increasing concentrations of DARA (Fig S1C). Upon the addition of an IgG1 control antibody (anti-IgE, referred to as omalizumab [OMA]), we found no effect on B-cell proliferation after activation with TI or TD stimuli compared with the conditions where no mAb was added (Fig S2A and B). Thus, targeting CD38 with DARA reduced B-cell proliferation upon TD stimulation.

## SLAMF7 suitable as a substitute for CD38 as a PB marker

Next, we studied the phenotypic and functional characteristics of B-cell differentiation. As mentioned, CD38 has extensively been used to classify PBs. Because DARA interferes with conventional fluorescently labeled anti-CD38 mAbs as a result of targeting similar epitopes, an alternative marker was required to investigate PB formation in the presence of DARA (Fig 2A) ([32], [33]). The up-regulation of signaling lymphocyte activation marker family member 7 (SLAMF7), C-X-C chemokine receptor type 4 (CXCR4), and B-cell maturation antigen (BCMA), and the down-regulation of B-cell activation factor receptor (BAFFR) were identified as possible alternative extracellular markers for PB identification ([34], [35], [36]). Proliferating B cells (CFSE$^{low}$) were BAFFR-negative, and SLAMF7- and CXCR4-positive at day 6 (Fig S3A). BCMA expression could only be detected if γ-secretase was added at the start of culture to prevent cleavage (data not shown) ([37]). To find a sufficient replacement for CD38 to phenotypically identify PB, the CD27$^+$CD38$^+$ population was compared with CD27$^+$SLAMF7$^+$, CD27$^+$CXCR4$^+$, CD27$^+$BCMA$^+$, and CD20$^-$BAFFR$^-$ population after 6 d of in vitro culture. In addition, the CFSE$^{low}$CD38$^+$ population was compared with the CFSE$^{low}$SLAMF7$^+$, CFSE$^{low}$CXCR4$^+$, CFSE$^{low}$BCMA$^+$, and CFSE$^{low}$BAFFR$^-$ populations (Fig S3B–E). SLAMF7 expression showed consistent similarity with CD38 expression, especially for TD stimulations (Fig S3B), whereas the other markers did not. To determine the resemblance of CD38 and SLAMF7 expression over time in the B-cell differentiation assay, the expression of CD38 and SLAMF7 was measured after 3, 4, 5, and 6 d of in vitro culture after stimulation with TD and TI stimuli. The development of PBs determined either by CD27$^+$CD38$^+$ or by CFSE$^{low}$CD38$^+$ compared with CD27$^+$SLAMF7$^+$ or CFSE$^{low}$SLAMF7$^+$ showed very similar kinetics during culture (Fig S4A and B). Thus, PBs developed over time, which is further confirmed by the gradual induction of IgG, IgA, and IgM production and release into the supernatant over time (Fig S4C). We found no up-regulation of CD38 under these stimulations in T cells (Fig S4D). In conclusion, SLAMF7 is a suitable alternative marker for CD38 in regard to B-cell differentiation. In further experiments, SLAMF7 was used to identify PBs when CD38 was insufficient to use.

## Daratumumab limits in vitro B-cell differentiation and IgG production

To assess the effect of DARA on B-cell differentiation, we analyzed the formation of PBs (CD27$^+$SLAMF7$^+$, CFSE$^{low}$SLAMF7$^+$) and PCs (CFSE$^{low}$CD138$^+$) after 6 d of culture and supernatants were collected to measure immunoglobulin production. Dose–response kinetics

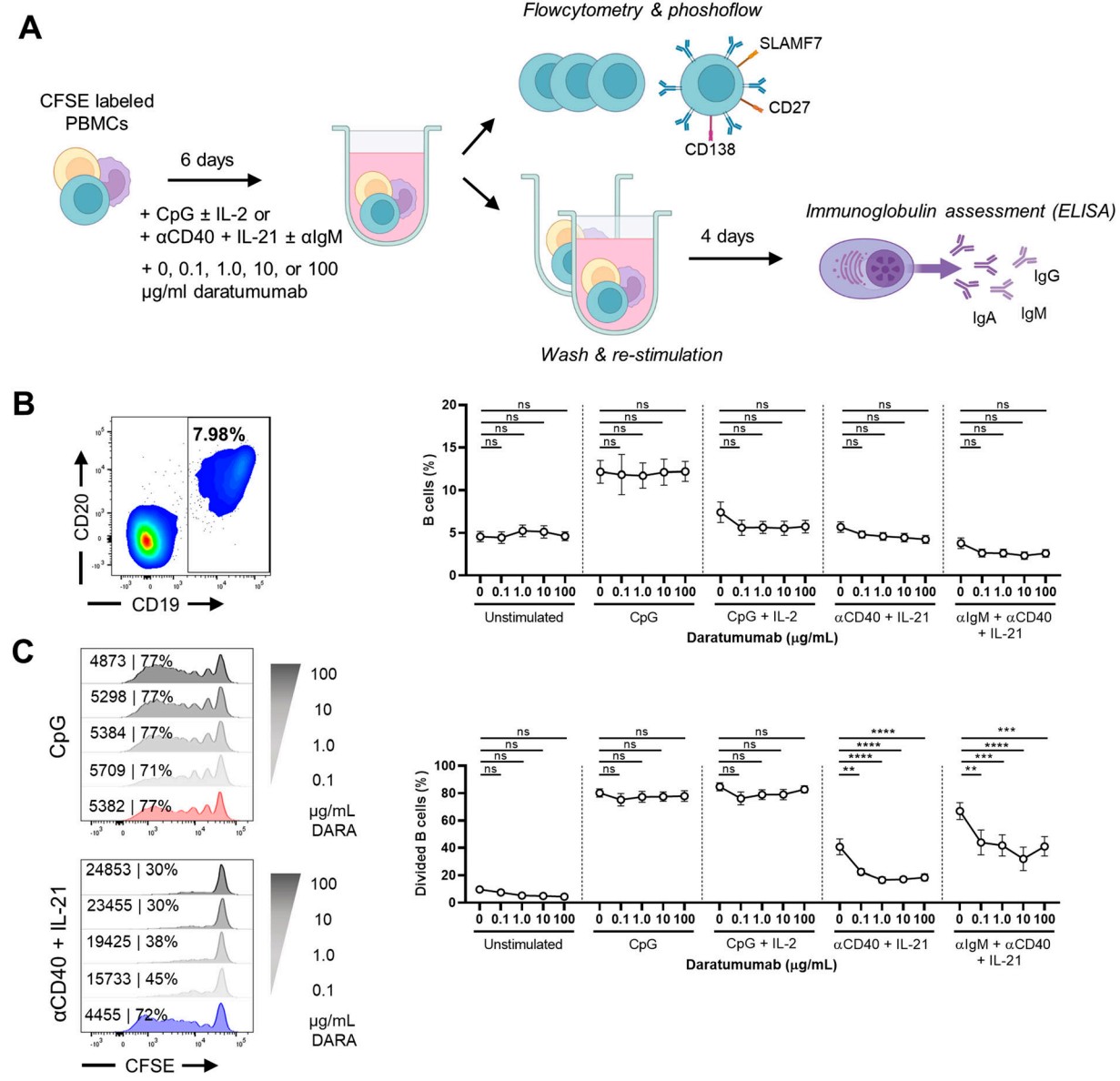

**Figure 1. Characterization of an in vitro system to establish the effects of daratumumab on B-cell proliferation.**
**(A)** Schematic overview of the experimental setup (created with BioRender.com). In short, CFSE-labeled PBMCs were cultured for 6 d without or with various concentrations of daratumumab and stimulated with CpG ± IL-2 or αCD40 + IL-21 ± αIgM. For some experiments, cells were washed at day 6 and restimulated until day 10. **(B)** Representative CD19/CD20 FACS plot (left) after 6 d of culture with CpG showing the gating of CD19$^+$ B cells. Quantification (right) of the percentages CD19$^+$ B cells (of lymphocytes) for the different conditions tested. n = 6–8. **(C)** Amount of proliferation by CFSE dilution. Representative histogram overlays of CD19$^+$ B cells (left) for CpG and αCD40 + IL-21 stimulation at day 6. Values depicted next to the histograms represent the corresponding geometric MFI and the percentages of divided B cells. Quantification (right) of the percentages of divided CD19$^+$ B cells for the different conditions tested. n = 6–8. *P*-values were calculated using a one-way ANOVA and Dunnett's multiple comparisons test for each condition. ns, not significant, **P ≤ 0.01, ***P ≤ 0.001, and ****P ≤ 0.0001. Means ± SEM are displayed.

revealed >50% decrease in the percentages of CD27$^+$SLAMF7$^+$ and CFSE$^{low}$SLAMF7$^+$ PBs and CFSE$^{low}$CD138$^+$ PCs already in the presence of the lowest concentration of DARA in the cultures stimulated with CpG, αCD40 + IL-21, and αIgM + αCD40 + IL-21 (Figs 2B and C and S5A). When PBMCs were cultured in the presence of the IgG1 control antibody OMA, no significant effect was found on CD27$^+$SLAMF7$^+$ and CFSE$^{low}$SLAMF7$^+$ PB formation (Fig S5B and C). In addition, DARA and OMA showed no effect on the proliferation and up-regulation of the activation marker CD25 upon stimulation with αCD3 + αCD28

measured in the CD4$^+$ and CD8$^+$ T cells (Fig S5E and F). The exact measurement of IgG secretion in these culture supernatants was challenging, because DARA is a human IgG and interferes with the IgG ELISA (data not shown) (38). Therefore, PBMCs that were cultured with DARA for 6 d were washed and then restimulated in fresh medium with the same combination of stimuli for another 4 d. Supernatants of the additional 4-d culture (days 6–10) were collected and analyzed. Interestingly, we found a significant reduction in IgG production when B cells were exposed to different

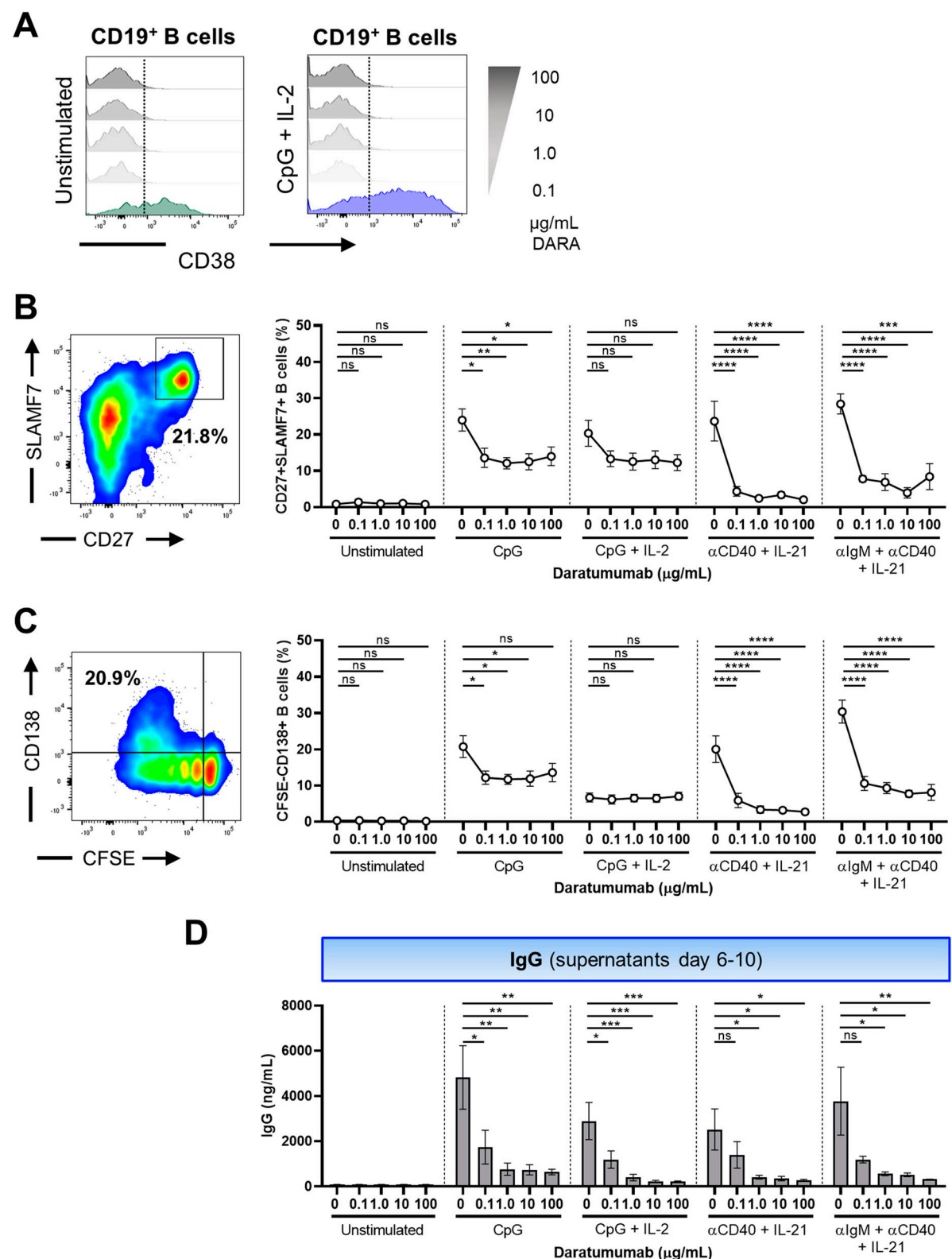

**Figure 2. Effect of daratumumab on B-cell differentiation and IgG production.**
(A) Representative histogram overlays of the CD38 expression of CD19[+] B cells in the presence of different concentrations of daratumumab. (B, C) Representative CD27/SLAMF7 FACS plot and CFSE/CD138 FACS plot (left) showing the gating of SLAMF7[+]CD27[+] and CFSE[low]CD138[+] B cells within the CD19[+] gate after stimulation with CpG. Quantification (right) of %SLAMF7[+]CD27[+] and %CFSE[low]CD138[+] B cells after 6 d of culture for the different conditions tested. n = 6–8. (D) IgG secretion measured in culture

concentrations of DARA in all conditions tested preceding the restimulation (Fig 2D). No significant effect of DARA on IgA and IgM secretion could be detected in the culture supernatants between days 0 and 6 and between days 6 and 10 (Fig S6A–D). Overall, these results suggest that targeting CD38 with DARA has an effect on B-cell differentiation and IgG production using TI and TD stimuli, whereas it did not affect T-cell proliferation or activation present in these cultures.

## Daratumumab attenuates activation-induced phosphorylation and acetylation of NF-κB in B cells

The extent of B-cell differentiation is regulated by a complex interplay of signaling pathways and transcription factors (TFs) (1). The canonical and non-canonical NF-κB pathway has been shown to play a critical role in B-cell development, activation, and survival (39). Classical NF-κB activation results after stimulus-coupled phosphorylation of cytoplasmic IκBα inhibitors by IκB kinases. These IκBα inhibitors are degraded by the proteasome allowing p50/p65 NF-κB complexes to enter the nucleus and stimulate target gene expression (40). For the full transcriptional activity of p65, it has been previously shown that phosphorylation at serine 529/536 and acetylation at lysine 310 of p65 are key post-translational modifications needed for the full transcriptional activity of NF-κB (Fig 3A) (41, 42, 43). The constitutive activation of p65 and the subsequent expression of target genes including BLIMP1 can drive PC formation. In an attempt to identify cellular processes affected by DARA treatment, we investigated the effect of DARA on p65 using phosphoflow cytometry (44). B cells were cultured with TI or TD stimuli in the absence or presence of DARA (10 μg/ml) and analyzed after 4 h or 3 d, as these time points were the most effective for tracking signaling events by phosphorylation and acetylation of NF-κB (44). To accommodate the differences in fluorescence intensity between donors, the ratio of unstimulated to stimulated B cells was calculated by normalizing to the expression in unstimulated (without DARA) B cells (set at a value of 1). Upon activation with TI or TD stimuli, we observed the reduced expression of IκBα in B cells already after 4 h of stimulation, but not in T cells, suggesting successful stimulus-coupled degradation of IκBα (Figs 3B and S7A and B). DARA treatment did not affect IκBα degradation or re-expression. At both time points, we also observed the increased expression of phosphorylated p-65 (p-p65), whereas after 3 d of stimulation, acetylated-p65 (Ac-p56) could be observed in B cells (Figs 3C and D and S7C and D). Interestingly, DARA treatment lowered the ratio of p-p65 induction upon TD stimulation and the ratio of Ac-p65 induction upon TI and TD stimulation after 3 d of culture. Total p65 protein levels were up-regulated after 3 d of stimulation, but remained unaffected by DARA treatment (Figs 3E and S7E). In the same experiments, DARA treatment had no effect on NF-κB protein levels and modifications in T cells (Fig S7B–E), co-inciding with the lack of CD38 up-regulation observed upon these B cell–specific stimulations (Fig S4D). We found no effect of DARA on phosphorylation of ERK 1/2, downstream of the BCR and TLR9, or

phosphorylated or total levels of STAT3, downstream of the IL-21 receptor (Fig S8A–C). In addition, we measured the expression of PAX5, BLIMP1, and IRF4 as indicators for their transcriptional status. Generally, strong NF-κB signaling allows up-regulation of IRF4 and the expression of BLIMP1 while down-regulating PAX5, together resulting in PC differentiation (Fig 4A) (4, 5). B cells were stimulated as above in the absence or presence of two concentrations of DARA (0.1 and 10 μg/ml). B cells were analyzed on days 3 and 6, as these time points were the most effective for tracking the induction of differentiation as we have shown that CD27⁺SLAMF7⁺ PBs can be detected from day 3 onward using these stimuli (Fig S4). BLIMP1 expression analysis showed the expected increase in the percentage of BLIMP1^high B cells over time upon CpG or αCD40 + IL-21 stimulation, and there was a downward trend upon the addition of different concentrations of DARA (Figs 4B and C and S8D). To examine the TF profile at day 6, the co-expression of these TFs was analyzed. Upon DARA treatment, >50% reduction in BLIMP1⁺IRF4⁺ B cells was observed in the CpG-stimulated conditions, with an expected increase in BLIMP1⁻IRF4⁻ B cells (Fig 4D). In addition, in the same conditions we found a small reduction in BLIMP1⁺PAX5⁻ B cells but, surprisingly, without an increase in the PAX5⁺BLIMP1⁻ B cells (Fig 4E). This indicates that DARA attenuates activation-induced phosphorylation and acetylation of NF-κB in B cells, which would normally lead to gene transcription and the up-regulation of BLIMP1.

## Daratumumab inhibits the proliferation and differentiation of sorted CD19⁺IgD⁻CD27⁺ memory B cells but not T cells

Because the starting population of B cells comprises both naïve, non-switched, and switched memory B cells, we investigated which of these subsets was most affected by DARA. FACS-sorted naïve (CD19⁺IgD⁺CD27⁻), non-switched memory (CD19⁺IgD⁺CD27⁺), and switched memory (CD19⁺IgD⁻CD27⁺) B-cell populations with the addition of autologous non-B cells (mostly T cells) were cultured with or without DARA and stimulated with CpG or αCD40 + IL-21 for 6 d (Fig S9A and B). In this system, sorted switched memory B cells generally differentiate stronger to TI and TD stimulation than the naïve and non-switched B cells (Fig 5B). Again, we found no effect of DARA on the B-cell percentages using the sorted B-cell fractions (Fig S9C). The addition of DARA caused significantly reduced proliferation and differentiation of memory B cells with αCD40 + IL-21 (Figs 5A and B and S9D), while only showing a non-significant trend upon CpG stimulation. Lastly, immunoglobulin production of the memory B-cell fraction stimulated with αCD40 + IL-21 showed a significant decrease in IgG and IgA, but not IgM production (being hardly detectable at all under these conditions) (Fig S10A–C). Because of the previously mentioned interference of DARA with the IgG ELISA, we report here an estimated-IgG production where the levels were corrected for DARA present in the culture supernatant. Naïve B cells produced low amounts of IgM independently of DARA treatment, whereas non-switched B cells produced large quantities of IgM and low quantities of IgA independent of treatment with

---

supernatants between days 6 and 10 (after washing and restimulation). n = 4. *P*-values were calculated using a one-way ANOVA and Dunnett's multiple comparisons test for each condition. ns, not significant, *P ≤ 0.05, **P ≤ 0.01, and ***P ≤ 0.001. Means ± SEM are displayed.

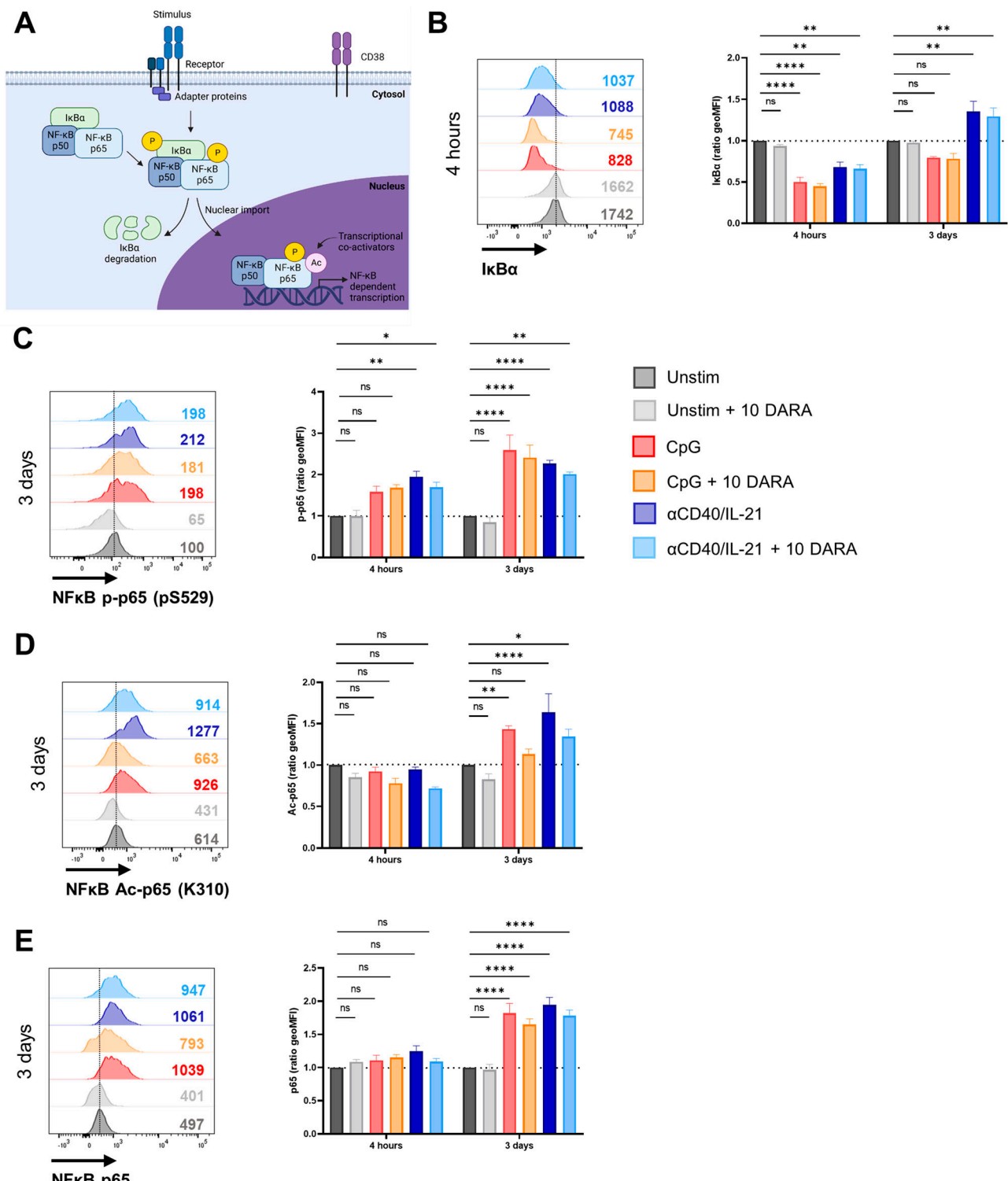

**Figure 3. Daratumumab attenuates activation-induced phosphorylation and acetylation of NF-κB in B cells.**
**(A)** Schematic representation of phosphorylation and acetylation events of NF-κB proteins upon stimulation. **(B)** Representative histogram overlays (left) of the IκBα expression of CD19[+] B cells after 4 h of stimulation with indicated stimuli ± 10 μg/ml daratumumab. The geometric mean (geoMFI) ratio (right) was calculated by normalizing to the expression in unstimulated CD19[+] B cells (set at a value of 1) at the corresponding time point. n = 3. **(C, D, E)** Representative histogram overlays (left) of (C) NF-κB phospho-p65 (pS529), (D) NF-κB acetylated-p65 (K310), and (E) NF-κB total-p65 and expression of CD19[+] B cells after 3 d of stimulation with indicated stimuli ± 10 μg/ml daratumumab. The geometric mean (geoMFI) ratio (right) was calculated by normalizing to the expression in unstimulated CD19[+] B cells (set at a value of 1) at the corresponding time point. n = 3. P-values were calculated using a two-way ANOVA and Dunnett's multiple comparisons test. ns, not significant, *$P \leq 0.05$, **$P \leq 0.01$, and ****$P \leq 0.0001$. Means ± SEM are displayed.

Figure 4. Effects of daratumumab on B-cell transcription factors during differentiation.
(A) Schematic representation of the expression of transcription factors PAX5, BLIMP1, and IRF4 during B-cell activation and differentiation (created with BioRender.com).
(B) Representative histogram overlays of the BLIMP1 expression of CD19⁺ B cells after 3 and 6 d of stimulation with indicated stimuli and daratumumab (0.1 or 10 µg/ml).
(C) Quantification of the percentages of BLIMP1ʰⁱᵍʰ B cells at day 6 for the different conditions tested. n = 3. (D) Representative BLIMP1/IRF4 FACS plots (left) showing the gating of BLIMP1⁻IRF4⁻ and BLIMP1⁺IRF4⁺ within the CD19⁺ gate after stimulation with CpG. Quantification (right) of %BLIMP1⁻IRF4⁻ (green) and %BLIMP1⁺IRF4⁺ (red) B

DARA. Unlike B cells, activation and proliferation of sorted CD3[+] T cells were not affected by DARA upon αCD3 + αCD28 stimulation determined by the CFSE dilution and an increase in the CD25 expression of CD4[+] and CD8[+] T cells (Fig 5C and D). In sum, primarily memory B cells stimulated with αCD40 + IL-21 were affected by DARA.

# Discussion

CD38 is a glycoprotein found on the surface of many (activated) immune cells, including CD4[+], CD8[+], B lymphocytes, and natural killer cells (45). Expression is highest on PBs and PCs, and the malignant counterpart in MM (46). Apart from its approval for MM treatment, CD38-targeting mAb DARA is gaining attraction as a rescue therapy for autoimmune conditions with PC involvement (Table S1). Hitherto, the promising evidence for DARA as a therapeutic option in autoimmune diseases is based exclusively on individual case reports (47). How anti-CD38 therapy influences normal PCs or PC differentiation beyond depletion in cancer settings has remained virtually unexplored. In this study, we observed profound differences in B-cell proliferation, differentiation, and immunoglobulin production under DARA treatment in vitro at very low doses already. Mechanistically, we show that DARA influences the induction of differentiation-associated signaling and TF levels in B cells. Furthermore, our data imply that the presence of DARA most prominently leads to inhibition of the switched memory B-cell responses, which could be beneficial in an autoimmune setting but disadvantageous upon infection or in vaccination strategies.

Upon the addition of DARA, the CD38 expression on B cells was unmeasurable with fluorescently labeled mAbs, which was reported previously as well (34). The suitability of the surface expression of SLAMF7 for the identification of PBs in peripheral blood confirms previous studies for bone marrow samples (34, 48). Proliferation of B cells only showed a significant decrease in αCD40 + IL-21 (±αIgM)– and not CpG-stimulated conditions, indicating that TLR9 stimulation seems independent of any CD38 influence in B-cell activation. In a recent study, human peripheral B cells were stimulated with Fab fragments (αIgM/G/A) + CpG + IL-2 in the presence of α-CD38 mAbs (clone HB-7 and DARA) showing reduced proliferation after 5 and 7 d (28), which was suggested to be causally explained by pERK protein expression at early time points in B cells pretreated with DARA (within 1 h). We were unable to confirm the effect of DARA on early ERK phosphorylation in our cultures. Although the combination of stimuli and the setup of experiments (short and long stimulations) differ from our study (28), collectively these data suggest that DARA inhibits B-cell proliferation upon either BCR or CD40, but not TLR9 stimulation, coinciding with perturbations in protein phosphorylation and acetylation.

In our 6-d B-cell differentiation assay, we observed an effect of DARA on B-cell differentiation and IgG production irrespective of TD or TI stimulation. For elucidation of cues promoting B-cell differentiation into PCs, here it is important to highlight that naïve (IgD[+]CD27[−]), non-switched memory (IgD[+]CD27[+]), and switched memory (IgD[−]CD27[+]) B cells react differently upon TD and TI signals. When total B cells are stimulated with CpG, B cells differentiate into CD27[+]SLAMF7[+] PBs and secrete IgG, IgA, and IgM immunoglobulins. We show that stimulation of total B cells with CpG essentially measures the function of switched memory B cells (29), as they produce the vast majority of IgG and IgA, whereas the non-switched memory B cells produce large quantities of IgM in vitro. The production of IgM was not reduced in both purified and non-purified cultures, indicating that the non-switched B-cell populations were unaffected. Upon CpG stimulation, a consistent trend was observed in reduced differentiation and IgG production upon CpG stimulation, either in the total B-cell or in the sorted B-cell subset cultures. In addition, we found reduced PCs upon stimulation both on phenotypic and on transcriptional levels. An important finding was that the proliferation, differentiation, and IgG production upon TD stimulation with αCD40 + IL-21 were strongly inhibited by DARA, coinciding with reduced activation of NF-κB and PC TF expression. Moreover, IgA and IgG production by sorted memory B cells stimulated with αCD40 + IL-21 was strongly reduced at very low concentrations. A preceding in vivo study regarding the pharmacokinetics of DARA showed a serum concentration 10- to 100-fold higher than the active concentrations used in this study for several months (49). Although T cells express CD38 upon activation (45), both CD4[+] and CD8[+] T-cell activation and proliferation were not affected by DARA upon T-cell stimulation with αCD3/CD28.

The strength of this study is that we have investigated both TD and TI B-cell responses in the presence of DARA with a strong impact on TD stimulation and—in this respect—a differential outcome of different B-cell subsets, demonstrating the most impact of CD38 targeting on the activation and maturation of memory B cells into PBs and PCs. Although derived from an in vitro system, our findings do suggest a possible mechanism for the additional risk of bacterial and viral infections in MM patients receiving anti-CD38–directed therapies in vivo (50, 51). Where normally the barrier of immunological memory would protect patients, reduced recall responses by memory B cells upon TI or TD stimulation leave patients who receive DARA treatment prone to infections with common pathogens (16). As mentioned previously, patients treated thus far with DARA show reduced humoral responses to anti-SARS-CoV-2 mRNA vaccinations (52, 53, 54, 55), although this negative impact on vaccinations was less obvious in other studies (40, 56). Although these mixed observations suggest that DARA treatment may negatively affect humoral responses in the context of vaccination, this needs further confirmation. Our data showed that T-cell activation was not affected by DARA treatment, suggesting that cellular responses by T cells would not be affected by DARA

cells after 6 d of culture for the different conditions tested. n = 3. **(E)** Representative BLIMP1/PAX5 FACS plots (left) showing the gating of PAX5[+]BLIMP1[−] and PAX5[−]BLIMP1[+] within the CD19[+] gate after stimulation with CpG. Quantification (right) of %PAX5[+]BLIMP1[−] (orange) and %PAX5[−]BLIMP1[+] (purple) B cells after 6 d of culture for the different conditions tested. n = 3. *P*-values were calculated using a two-way ANOVA and Dunnett's multiple comparisons test. ns, not significant, *$P \leq 0.05$, and **$P \leq 0.01$. Means ± SEM are displayed.

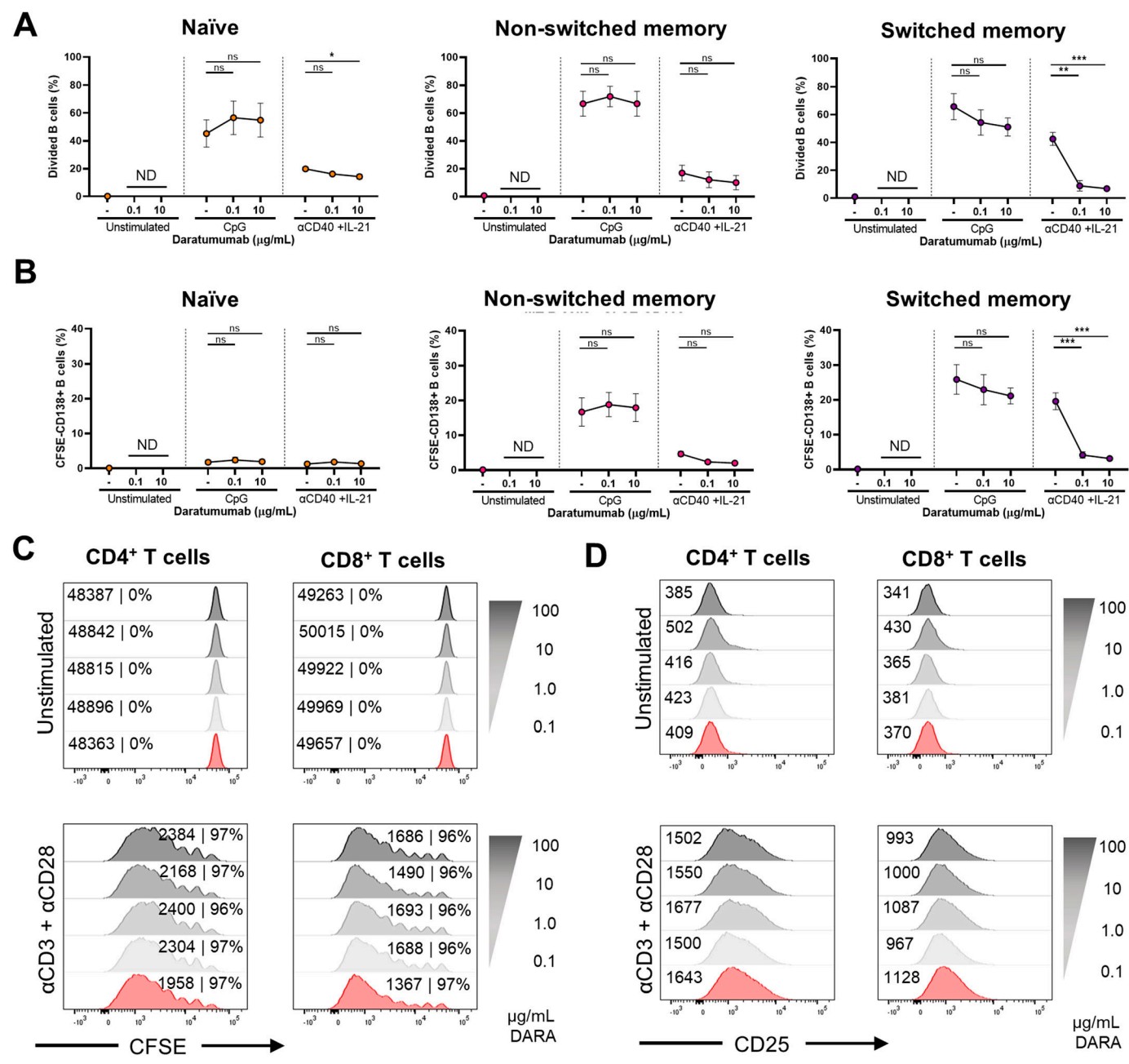

**Figure 5. Effects of daratumumab on sorted naïve and memory B-cell subsets and T cells.**
**(A, B)** FACS-sorted naïve (CD19⁺IgD⁺CD27⁻), non-switched (CD19⁺IgD⁺CD27⁺), and memory (CD19⁺IgD⁻CD27⁺) B-cell populations were co-cultured with autologous non-B cells with CpG or αCD40 + IL-21 and daratumumab (0.1 or 10 μg/ml) for 6 d. **(A, B)** Quantification of the percentages of (A) divided and (B) CFSE⁻CD138⁺ B cells at day 6. n = 3.
**(C, D)** FACS-sorted CD3⁺ T cells were cultured with αCD3+αCD28 and five different concentrations of daratumumab for 6 d. **(C, D)** Amount of proliferation by CFSE dilution and (D) amount of activation by CD25 up-regulation. Representative histogram overlays of CD4⁺ (left) and CD8⁺ (right) T cells at day 6 after no stimulation or αCD3 + αCD28. Values depicted next to the histograms represent the corresponding geometric MFI and the percentages of divided CD4⁺ and CD8⁺ T cells, respectively. n = 3. *P*-values were calculated using a two-way ANOVA and Dunnett's multiple comparisons test. ns, not significant, *P ≤ 0.05, **P ≤ 0.01, and ***P ≤ 0.001. Means ± SEM are displayed.

treatment after vaccination and would therefore provide additional protection.

Although the mode of action remains to be fully explained, there are several explanations possible. First, targeting CD38 with an antibody or removing this molecule with CRISPR/Cas9 has been reported to inhibit the association of CD19 with the IgM-BCR, directly impairing BCR signaling in normal and malignant B cells (28). Our data that DARA had little if any effect with naïve and non-switched memory B cells upon BCR stimulation are not easily reconciled with this explanation as the only mode of action. Alternatively, CD38 acts as a receptor and an ectoenzyme (45) and gained appreciation as an immune metabolic modulator in

multiple immune cells ([57], [58], [59]). The enzymatic functions attributed to CD38 are primarily investigated in mouse and tumor models ([16], [60]). Clinical relevance is sought in the relationship between the membrane expression of CD38 and the extracellular and cytoplasmic $NAD^+$ levels. Engagement of CD38 using mAbs such as daratumumab or isatuximab would block the enzymatic activities ([61]), further supported by in vitro studies with MM cell lines ([62]). In the extracellular compartment, it would potentially contribute to the generation of adenosine monophosphate and adenosine by the ectoenzyme CD38/CD203a/CD73 pathway, which would then suppress B and T cells ([60], [63], [64]). Although not excluded at the biochemical levels, in our studies DARA did not inhibit T-cell activation indicating a differential effect if present. In the intracellular compartment, increased $NAD^+$ levels could influence the subsequent activity of several $NAD^+$-consuming enzymes, which could play a role in orchestrating fate decisions, in particular in those cells with increased expression levels such as PCs ([60]). Key $NAD^+$-consuming enzymes are poly(ADP-ribose) polymerase-1/2 (PARP-1/2) and the family of sirtuins (SIRTs), which play important roles in cell death, aging, and metabolic regulation of immune cell function ([65], [66]), including B cells ([65], [67]). More importantly, these proteins can modulate transcriptional co-activators and thereby influence TF activity. In knockout mice, CD38 deficiency has been associated with increased activation of the NAD-dependent deacetylating enzyme SIRT1 ([24], [25]), which also deacetylates acetyllysine in selected proteins involved in DNA transcription. By deacetylating NF-κB p65 at lysine 310, SIRT1 also inactivates NF-κB ([25]). This may correspond to the role of SIRT1 to modulate activation-induced cytidine deaminase (gene symbol AICDA, protein abbreviated AID) expression, a protein involved in class-switch recombination, and the antibody response in human B cells ([68]). These data would match the lack of an inhibitory effect of DARA on non-switched IgM-producing B cells shown here. Our demonstration of reduced phosphorylation and acetylation of NF-κB through direct or indirect effects of DARA on CD38 function during stimulation would outline a possible mechanism by which the full transcriptional activity of NF-κB is abolished, resulting in less PB and PC formation and IgG secretion in vitro.

In conclusion, our study has identified a mechanism to explain why MM patients receiving DARA show reduced autoantibody levels, lower vaccination responses, and increased infection risks ([12], [15], [69]). We showed that DARA inhibits proliferation and differentiation of B cells into PBs and PCs, although the function of CD38 needs to be elucidated to obtain further knowledge about the exact mechanism of action of DARA. With the expected increase in the use of DARA therapeutically and current knowledge, it would be reasonable to consider infectious prophylaxis in patients receiving DARA to counteract the possible increased risk of infections and/or vaccine failures.

# Materials and Methods

## Samples

PBMCs were derived from buffy coats from healthy donors. All the healthy donors provided written informed consent in accordance with the protocol of the local institutional review board, the Medical Ethics Committee of Sanquin Blood Supply, and the study conformed to the principles of the Declaration of Helsinki. PBMCs were isolated using density gradient centrifugation with Lymphoprep (Serumwerk Bernburg). After isolation, PBMCs were cryopreserved and stored in liquid nitrogen until further use.

## Human B-cell cultures and proliferation assay

First, PBMCs were thawed and resuspended in PBS at a concentration of $5-10 \times 10^6$ cells/ml. Cells were labeled with 0.5 μM CFSE (FITC; Molecular Probes) for 12 min at 37°C under constant agitation. The labeled PBMCs were washed with PBS and resuspended in IMDM supplemented with 10% FCS (BioWhittaker), 0.05 mg/ml gentamicin (Gibco), and $3.6 \times 10^{-4}$% v/v β-mercaptoethanol (Merck). CFSE-labeled PBMCs containing a fixed number of $2.5 \times 10^4$ B cells were cultured in 96-well U-bottom plates for 4 h, 3 d, or 6 d at 37°C and stimulated with 1 μg/ml CpG oligodeoxynucleotide 2006 (InvivoGen) ± 100 U/ml IL-2 (R&D Systems), or with 1 μg/ml αCD40 mAb (clone 14G7; Sanquin) and 20 ng/ml IL-21 (Invitrogen) ± 5 μg/ml αIgM mAb (clone MH15; Sanquin). For T-cell stimulation, cells were stimulated with saturating amounts of soluble αCD3 (clone 1xE) and αCD28 (cone 15E8). PBMCs were cultured in the presence of various concentrations of daratumumab (Darzalex; Janssen Pharmaceuticals) (0, 0.1, 1.0, 10, or 100 μg/ml). In some experiments, PBMCs were cultured with various concentrations of mAb OMA (anti-IgE, human IgG1 kappa; Novartis) (1.0 or 10 μg/ml). For specific experiments that required analysis of BCMA, samples were cultured with 100 nM γ-secretase inhibitor to prevent cleavage of BCMA. Supernatants were collected for further analysis. To obtain supernatants that did not contain daratumumab, in specific experiments, cells were washed after day 6 and restimulated for 4 d with stimuli as described above, after which supernatants were collected.

## Flow cytometry

### Extracellular staining of surface markers
PBMCs were resuspended in PBS supplemented with 0.5% wt/vol BSA, 2 mM EDTA, and 0.01% sodium azide and incubated with saturating concentrations of fluorescently labeled conjugated mAbs for 30 min at 4°C under constant agitation as described before ([29]). Cells were analyzed using a FACSCanto II flow cytometer and FACSDiva software (BD Biosciences). Using FlowJo software, proliferation of B and T cells was determined by measuring CFSE dilution, whereas activation and differentiation were assessed by the expression of CD25, CD27, CD38, CD138, and SLAMF7. The following mAbs (indicated as panel 1) were used for immunophenotyping: CD3 APC-R700 (557943; BD Biosciences), CD4 PE-Cy7 (348809; BD Biosciences), CD8 PerCP-Cy5.5 (341050; BD Biosciences), CD19 APC-R700 (564977; BD Biosciences), CD20 PerCP-Cy5.5 (332781; BD Biosciences), CD25 APC (340907; BD Biosciences), CD27 APC (337169; BD Biosciences), CD27 APC-eFluor 780 (47-0279-42; eBioscience), CD38 PE (345806; BD Biosciences), CD38 PE-Cy7 (335825; BD Biosciences), CD138 APC (347216; BD Biosciences), BAFFR APC (316916; BioLegend), BCMA PE (357504; BioLegend), CXCR4 PE-Cy7 (306514; BioLegend), IgD PE (555779; BD Biosciences), and SLAMF7 PE-Cy7 (331815; BioLegend).

### Intracellular staining of signaling and TFs

Cells were harvested, pooled, and pelleted before washing twice with 10 ml of PBS/0.1% BSA. Next, cells were fixed and stained according to previously published protocols for intracellular stainings and TF-flow (44). In short, the (1) intracellular staining protocol uses the Fix/Perm kit from BD Biosciences according to the manufacturer's instructions (Cytofix 554655; Perm. Buffer 558050) and the (2) TF staining protocol uses the Foxp3 fixation buffer (eBioscience, through Thermo Fisher Scientific) and Foxp3 permeabilization buffer (eBioscience). (1) For the intracellular staining, cells were extracellularly stained for 20 min at 4°C with saturating concentrations of CD19 APC-R700 (564977; BD Biosciences). Labeled cells were fixed and permeabilized using the Fix/Perm kit from BD Biosciences. Cell pellets were divided into two separate stainings (indicated as panels 2 and 3). Next, cells were stained intracellularly in PBA with saturating concentrations of the following fluorescently labeled mAbs: CD3 BV421 (562426; BD Biosciences), pERK 1/2 (T202/Y204) (560115; BD Phosflow), NF-κB p65 PE (653003; BioLegend), NF-κB phospho-p65 (S529) PE-Cy7 (560335; BD Biosciences), IκBα Alexa 647 (8993S; Cell Signaling), or NF-κB acetyl-p65 (k310) (ab19870; Abcam), and in combination with a secondary Ab in APC (709-136-149; Jackson ImmunoResearch) after incubation and washing. (2) For the TF staining (indicated as panel 4), cells were stained with 1:1,000 LIVE/DEAD Fixable Near-IR Dead Cell Stain Kit and CD19 BV510 (562947; BD Biosciences), SLAMF7 PE-Cy7 (331815; BioLegend), and CD38 V450 (646851; BD Biosciences) with or without CD27 BUV395 (563815; BD Biosciences) antibodies. Cells were incubated for 15 min in the fridge. Next, cells were washed and fixed using Foxp3 permeabilization buffer (eBioscience) on ice. After a washing step with this buffer, samples were stained with 25 μl of staining mix containing anti-PAX5 PE (649708; BioLegend), IRF4-PerCP-Cy5-5 (646415; BioLegend), anti-BLIMP1 AF647 (IC36081R-025; R&D Systems), pSTAT3 (612569; BD Biosciences), and tSTAT3 (564133; BD Biosciences) diluted in Foxp3 permeabilization buffer and incubated for 30 min in the fridge. The samples were washed and resuspended in a volume and measured on a BD FACSymphony A3 or A5 machine. The flow cytometer was calibrated by compensating for all conjugates using UltraComp eBeads Compensation Beads (Invitrogen). The data were analyzed using FlowJo software, v10.6.2 (Treestar).

### FACS

In separate experiments, CFSE-labeled PBMCs were resuspended in a culture medium and incubated with saturating concentrations of dye-conjugated mAbs for 30 min at 4°C. Naïve (CD20$^+$CD19$^+$IgD$^+$CD27$^-$), non-switched (CD20$^+$CD19$^+$IgD$^+$CD27$^+$), and memory (CD20$^+$CD19$^+$IgD$^-$CD27$^+$) B-cell populations and non–B-cell populations (CD19$^-$) were isolated by FACS with a FACSAria II (BD Biosciences) dependent on the experiment. In addition, from the non–B-cell fraction, total CD3$^+$ T cells (CD3$^+$CD19$^-$) were isolated for specific experiments. The following mAbs were used for isolation: CD3 PE (347247; BD Biosciences), CD19 APC-R700 (564977; BD Biosciences), CD27 APC (337169; BD Biosciences), and IgD PE (555779; BD Biosciences). Different combinations, at a fixed number of cells (25,000 B cells with 125,000 non-B cells) or 100,000 T cells, were then cultured in 96-well U-bottom plates for 6 d. During culture, the cells were stimulated with CpG or αCD40 + IL-21 with or without concentrations of daratumumab as described above.

T cells were stimulated with anti-CD3 (αCD3) (clone 1xE; Sanquin) and 10 μg/ml anti-CD28 (αCD28) (clone 15E8; Sanquin) with or without daratumumab. After 6 d, cells were analyzed by flow cytometry as described above. Supernatants were collected for further analysis.

### IgG, IgA, and IgM ELISA

The secretion of IgG, IgA, and IgM by cultured B cells was assessed with an in-house ELISA protocol using polyclonal rabbit anti-human IgG, IgA, and IgM, and a serum protein calibrator (all from Dako) as described previously (29).

### Statistics

Differences between the relative expression of markers and the secretion of immunoglobulins were calculated with a one-way or two-way ANOVA and Dunnett's multiple comparisons test where indicated. A $P$-value ≤ 0.05 was considered statistically significant.

## Supplementary Information

## Acknowledgements

We are thankful to the healthy donors that donated blood. We thank the Amsterdam UMC Pharmacy for the provision of left-over Darzalex and Xolair. We acknowledge the support of patient partners, private partners, and active colleagues of the T2B consortium (see website: www.target-to-b.nl). We thank the Sanquin Central Facility and the Laboratory Medical Immunology Amsterdam UMC for the maintenance and calibration of the FACS machines. This collaboration project is financed by the PPP Allowance made available by Top Sector Life Sciences and Health to Samenwerkende Gezondheidsfondsen (SGF) under project number LSHM18055-SGF to stimulate public–private partnerships and co-financing by health foundations that are part of the SGF.

### Author Contributions

D Verhoeven: conceptualization, data curation, formal analysis, investigation, visualization, methodology, and writing—original draft, review, and editing.
L Grinwis: conceptualization, data curation, formal analysis, investigation, visualization, and writing—original draft.
C Marsman: formal analysis, investigation, methodology, and writing—review and editing.
MH Jansen: data curation, formal analysis, and investigation.
EMM Van Leeuwen: conceptualization, supervision, and writing—original draft, review, and editing.
TW Kuijpers: conceptualization, supervision, funding acquisition, investigation, methodology, and writing—original draft, review, and editing.

## Conflict of Interest Statement

The authors declare that they have no conflict of interest.

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

# Appendix: T2B Consortium Members.

| PhD/Postdocs | Clinical partners |
|---|---|
| **Name, Institute** | **Name, Institute** |
| Mrs. Annabel Ruiter, LUMC | Dr. Filip Eftimov, AMC |
| Mrs. Linda van der Weele, AMC | Dr. Casper Franssen, UMCG |
| Mrs. Karoline Kielbassa, AMC | Prof. Dr. Jaap Groothoff, AMC |
| Mrs. Mariateresa Coppola, VUMC | Prof. Dr. Bart Jacobs Erasmus, MC |
| Mrs. Dorit Verhoeven, AMC | Dr. Barbara Horvath, UMCG |
| Mrs. Jyaysi Desai, LUMC | Prof. Dr. Arnon Kater, AMC |
| Mrs. Mirjam van der Burg, LUMC | Dr. Joep Killestein, VUMC |
| Mrs. Esther Vletter, LUMC | Prof. Dr. Taco Kuijpers, AMC |
| Mrs. Maaike Braham, RIVM | Dr. Karina de Leeuw, UMCG |
| Mr. Matthias Busch, MUMC | Dr. L Oosten, LUMC |
| Mr. Carlo Bonasia, UMCG | Dr. Pieter van Paassen, UMC Maastricht |
| Mrs. Elisabeth Raveling, UMCG | Dr. Bram Rutgers, UMCG |
| Mrs. Ruth Huizinga, ErasmusMC | Dr. Uli Scherer, LUMC |
| Mr. Niels Verstegen, Sanquin | Dr. Maarten Titulaer, ErasmusMC |
| Mr. Casper Marsman, Sanquin | Prof. Dr. Jan Verschuuren, LUMC |
| Mrs. Sabrina Pollastro, Sanquin | Dr. Niek de Vries, AMC |
| Mr. Koos van Dam, AMC | Dr. Diane van der Woude, LUMC |
| Mr. Laurent Paardekooper, LUMC | Dr. Josephine Vos, AMC |
| Mrs. Renée Ysermans, MUMC | Prof. Dr. Hendrik Veelken, LUMC |
| Mrs. Odilia Corneth, ErasmusMC | |
| Mrs. Annemarie Buisman, RIVM | |
| Mr. Rob van Binnendijk, RIVM | |
| Mrs. Pauline van Schouwenburg, LUMC | |
| Mr. Marvyn Koning, LUMC | |
| Mr. Luuk Wieske, AMC | |
| **Fundamental partners (group leaders/division heads)** | |
| Dr. Lisa van Baarsen, AMC | |
| Prof. Dr. Nico Bos, UMCG | |
| Dr. Anja ten Brinke, Sanquin | |
| Dr. Eric Eldering, AMC | |
| Dr. Cecile van Els, RIVM | |
| Prof. Dr. Marieke van Ham, Sanquin | |
| Prof. Dr. Peter Heeringa, UMCG | |
| Prof. Dr. Rudi Hendriks, ErasmusMC | |
| Dr. Maartje Huijbers, LUMC | |
| Dr. Ruth Huizinga, ErasmusMC | |
| Prof. Dr. Reina Mebius, VUMC | |

**Continued**

| PhD/Postdocs | Clinical partners |
|---|---|
| **Name, Institute** | **Name, Institute** |
| Dr. Theo Rispens, Sanquin | |
| Prof. Dr. Rene Toes, LUMC | |
| Dr. Jelle de Wit, RIVM | |
| Dr. Jan Damoiseaux, MUMC | |
| Dr. Wayel Abdulahad, UMCG | |

