## [Reviewer comments · Life Science Alliance]

Life Science Alliance

B cell targeting with anti-CD38 daratumumab: implications for differentiation and memory responses

Dorit Verhoeven, Lucas Grinwis, Casper Marsman, Machiel Jansen, Ester van Leeuwen, and Taco Kuijpers

DOI: <https://doi.org/10.26508/lsa.202302214>

Corresponding author(s): Dorit Verhoeven, Amsterdam University Medical Centers

Review Timeline:

Submission Date:	2023-06-13
Editorial Decision:	2023-06-16
Revision Received:	2023-06-16
Accepted:	2023-06-20

Transaction Report:

Please note that the manuscript was previously reviewed at another journal and the reports were taken into account in the decision-making process at *Life Science Alliance*. Since the original reviews are not subject to Life Science Alliance's transparent review process policy, the reports and author response cannot be published.

June 16, 2023

RE: Life Science Alliance Manuscript #LSA-2023-02214-T

Ms. Dorit Verhoeven
Amsterdam University Medical Centers
Experimental Immunology
Meibergdreef 9
Amsterdam 1105 AZ
Netherlands

Dear Dr. Verhoeven,

Thank you for submitting your revised manuscript entitled "B cell targeting with anti-CD38 daratumumab: implications for B cell differentiation and memory responses". We would be happy to publish your paper in Life Science Alliance pending final revisions necessary to meet our formatting guidelines.

- please upload all figure files as individual ones, including the supplementary figure files; all figure legends should only appear in the main manuscript file
- please add a Running Title, a Category, and a Summary Blurb/Alternate Abstract to our system
- please add the Twitter handle of your host institute/organization as well as your own or/and one of the authors in our system
- please consult our manuscript preparation guidelines <https://www.life-science-alliance.org/manuscript-prep> and make sure your manuscript sections are in the correct order, as well as what title page should include
- please add Author Contributions to the system, also
- please add your main and supplementary figure legends to the main manuscript text after the references section
- the Supplementary Methods and References should be incorporated into the main Methods and References sections, we have no limit on the size of these sections

A. FINAL FILES:

B. MANUSCRIPT ORGANIZATION AND FORMATTING:

Sincerely,

June 20, 2023

RE: Life Science Alliance Manuscript #LSA-2023-02214-TR

Ms. Dorit Verhoeven
Amsterdam University Medical Centers
Experimental Immunology
Meibergdreef 9
Amsterdam 1105 AZ
Netherlands

Dear Dr. Verhoeven,

Thank you for submitting your Research Article entitled "B cell targeting with anti-CD38 daratumumab: implications for differentiation and memory responses". It is a pleasure to let you know that your manuscript is now accepted for publication in Life Science Alliance. Congratulations on this interesting work.

DISTRIBUTION OF MATERIALS:

Again, congratulations on a very nice paper. I hope you found the review process to be constructive and are pleased with how the manuscript was handled editorially. We look forward to future exciting submissions from your lab.

Sincerely,
